# Strain and Strain Rate Tensor Mapping of Medial Gastrocnemius at Submaximal Isometric Contraction and Three Ankle Angles

Ryan Hernandez [1,†], Usha Sinha [1,†], Vadim Malis [2], Brandon Cunnane [1], Edward Smitaman [3] and Shantanu Sinha [2,*]

1   Department of Physics, San Diego State University, San Diego, CA 92182, USA
2   Muscle Imaging and Modeling Lab., Department of Radiology, University of California San Diego, San Diego, CA 92037, USA
3   Department of Radiology, University of California San Diego, San Diego, CA 92182, USA
*   Correspondence: shsinha@ucsd.edu; Tel.: +1-310-435-3994
†   These authors contributed equally to this work.

**Abstract:** Introduction: The aim of this study is to analyze the muscle kinematics of the medial gastrocnemius (MG) during submaximal isometric contractions and to explore the relationship between deformation and force generated at plantarflexed (PF), neutral (N) and dorsiflexed (DF) ankle angles. Method: Strain and Strain Rate (SR) tensors were calculated from velocity-encoded magnetic resonance phase-contrast images in six young men acquired during 25% and 50% Maximum Voluntary Contraction (MVC). Strain and SR indices as well as force normalized values were statistically analyzed using two-way repeated measures ANOVA for differences with force level and ankle angle. An exploratory analysis of differences between absolute values of longitudinal compressive strain ($E_{\lambda 1}$) and radial expansion strains ($E_{\lambda 2}$) and maximum shear strain ($E_{max}$) based on paired *t*-test was also performed for each ankle angle. Results: Compressive strains/SRs were significantly lower at 25%MVC. Normalized strains/SR were significantly different between %MVC and ankle angles with lowest values for DF. Absolute values of $E_{\lambda 2}$ and $E_{max}$ were significantly higher than $E_{\lambda 1}$ for DF suggesting higher deformation asymmetry and higher shear strain, respectively. Conclusions: In addition to the known optimum muscle fiber length, the study identified two potential new causes of increased force generation at dorsiflexion ankle angle, higher fiber cross-section deformation asymmetry and higher shear strains.

**Keywords:** muscle strain mapping; velocity-encoded MRI; fiber deformation asymmetry; shear strain

## 1. Introduction

Dynamic MRI studies using velocity-encoded phase-contrast imaging have enabled the extraction of 2D and 3D strain and strain rate tensors which provide information beyond one-dimensional strain measurements along the fiber [1–4]. The ability to measure both the compressive and radial expansion strains as well as shear strains enables a more detailed look at muscle and muscle fiber shape change during different types of contraction [5,6]. Further, principal strains can be extracted without the requirement for identifying the muscle fiber; this provides a lot of flexibility when muscle fibers are not visualized readily. Earlier studies on strain and strain rate tensor mapping have identified several features including, the anisotropy of deformation in the cross-section of the muscle fiber and deviation of the principal strain direction from the muscle fiber orientation [1,2,7]. There are several hypotheses and predictions from computational models that are related to some of these experimentally observed features [8]. For example, computational models have shown that the force output is increased when constraints to deformation are introduced in the fiber cross-section, i.e., a larger anisotropy of deformation in the fiber cross-section leads to larger forces being generated [8]. Computational models exploring force transmission from non-spanning fibers have identified that the shear in the endomysium can effectively

transmit the force to the tendon [9]. Regarding anisotropy of deformation and muscle shape changes, Eng et al. used a physical model based on an array of actuators to show that when the actuators contract against a load the actuators radially expand in the width direction but are prevented from expanding in the height direction. These constraints determine the final shape change of the muscle on contraction [5]. Most studies on shape changes monitor at the entire muscle level while the shape changes also occur at the muscle fiber level [6]. In contrast to ultrasound, VE-PC MRI and strain/SR tensor mapping also provides a convenient way to monitor shape changes in muscle fiber bundles (at the level of the voxel).

Muscle sarcomere force-length (FL) relationship is well established and describes the dependence of the steady-state isometric force of a muscle (or fiber, or sarcomere) as a function of muscle (fiber, sarcomere) length [10,11]. Muscle fiber architecture (fiber length and pennation angle) will clearly influence force production. Several researchers have examined the force produced by the Medial Gastrocnemius (MG) during isometric, concentric, and eccentric plantarflexion contraction for combinations of knee flexion and ankle positions [12–15]. The initial muscle fiber length and pennation angle of the MG changes with knee flexion and ankle angle. Earlier studies have used electromyography (EMG) and ultrasound (US) to study muscle isometric plantarflexion force, activation, and muscle architecture changes in the MG for combinations of knee flexion and ankle angles [13–15]. These studies are in general agreement that there is a decrease in force accompanied by a decrease in the activation of the MG at pronounced knee flexion positions, i.e., short muscle lengths.

Phase-contrast MR imaging has been successfully implemented to study muscle kinematics under different contraction paradigms as well as under different muscle conditions [1,3,4]. Strain describes how the tissue is deformed with respect to a reference state and requires tissue tracking. SR describes the rate of regional deformation and does not require 3D tracking or a reference state. A positive Strain or SR indicates a local expansion while a negative strain or SR indicates a local contraction. Strain and strain rate in the direction closest to the fiber provides information on the fiber contractility, while in the orthogonal directions it provides information about the deformation in the fiber cross-section allowing one to explore radial deformation and asymmetry of radial deformation. Another relevant index derived from the tensors is the shear strain and shear strain rate; these two indices may potentially reflect extent of lateral transmission of force [1,9]. The current paper focuses on analyzing the 2D strain and SR tensor during submaximal isometric contraction at two force levels and three ankle positions (dorsiflexed, neutral and plantarflexed) in order to extract principal (longitudinal compression and radial expansion) strains and shear strains. The hypothesis of the paper is that the dorsiflexed ankle position (i) will be the most efficient for force production, i.e., yield the smallest normalized compressive strain (normalized to force), (ii) have the highest anisotropy of deformation in the fiber cross-section, and (iii) largest shear strain compared to the longitudinal compressive strains. These three factors will together lead to a higher force generated at the dorsiflexed ankle angle compared to the neutral and plantarflexed angle.

## 2. Materials and Methods

A total of six male subjects (33.2 ± 16.3 yrs, height: 172.5 ± 7.0 cm, mass: 73.3 ± 6.5 kg) were included in this study; the criterion for inclusion was that subjects should be moderately active (defined as 150 to 300 min per week of moderate intensity activity such as brisk walking). The cohort included five young subjects (mean age: 26 years) and one old subject (66 years) so the current study lacked statistical power to look at age related differences; however, this will be the subject for future studies. Subjects participating in competitive sports or those with any surgical procedures on the lower leg were excluded. In addition, subjects were asked not to perform strenuous exercise during the preceding 24 h before the imaging session. The study was approved by the Medical Research Ethics Board of University of California San Diego and conformed to all standards for the use of human

subjects in research as outlined in the Declaration of Helsinki on the use of human subjects in research. Dynamic MR images were obtained of the subjects' lower dominant leg with a 1.5 T Signa HD16 MR scanner (General Electric Medical Systems, Milwaukee, WI, USA). The subjects were placed feet first in the supine position in the scanner and Figure 1 is a schematic that shows the positioning, visual feedback, force measurement and scan trigger setup. The dominant leg was placed into the foot pedal with the cardiac flex coil wrapped around the lower leg (the cardiac rather than a smaller coil was used in order to accommodate the foot pedal). The foot pedal device allowed for positioning and anchoring the foot at different ankle angles. In this study, the foot was positioned at three nominal angles—dorsiflexion (DF) 5°, neutral (N) −25°, and plantarflexion (PF) −40°. A large FOV image that included the ankle was collected at each foot position using the body coil to verify/estimate the ankle angle.

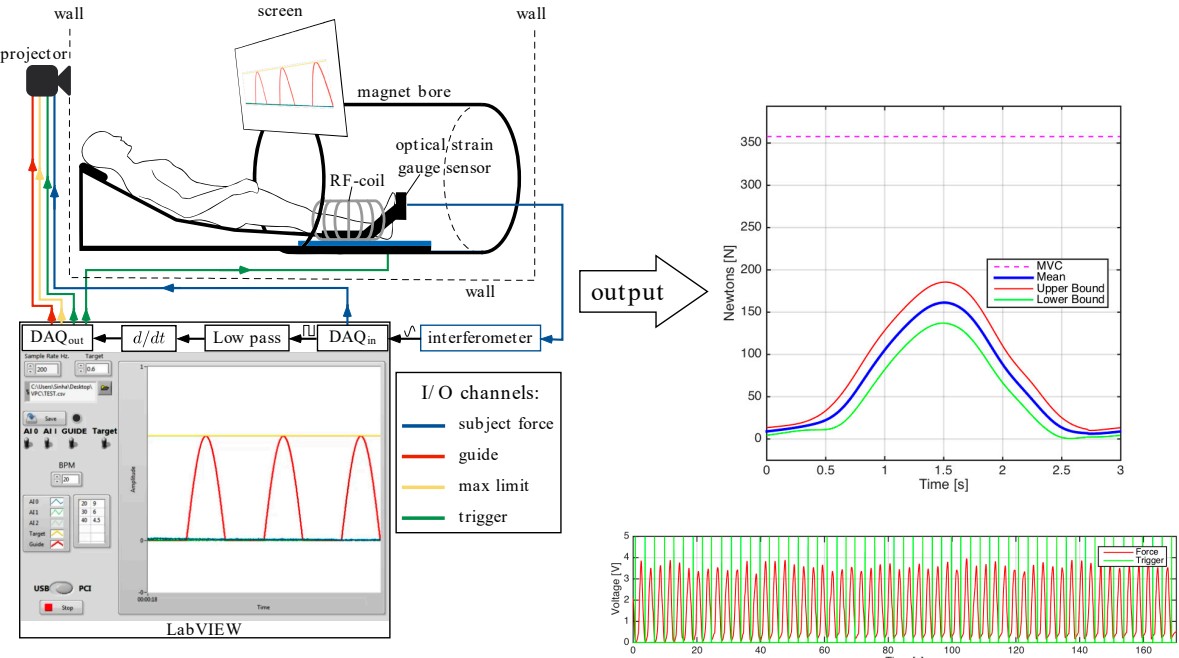

**Figure 1. Left top** panel: Subject setup with dominant leg in the foot pedal device, centered in a cardiac flex coil (labeled RF coil), with visual feedback projected onto the screen for the subject to follow. Pressure against the carbon-fiber plate in the foot pedal was detected by the transducer and converted into voltage and then converted into measurements of force. **Left bottom** panel: The foot pedal output was processed to generate a trigger to synchronize with the MR acquisition and also displayed to the subject on a screen. **Right top** panel: The force curve averaged over ~53 contractions (required to acquire the MR images) is shown along with upper and lower boundaries of the force curve. **Right lower** panel: Plot of the force curves for one VE-PC acquisition and the green vertical lines are the triggers. The foot positioning for neutral (N) ankle angle is shown here, dorsiflexed (DF) and plantarflexed (PF) positions were obtained by adjusting the ankle angle of the foot.

The ball of the foot rested on a carbon-fiber plate onto which an optical pressure transducer (Luna Innovations, Roanoke, VA, USA) was embedded (Figure 1). Pressure against the plate was detected by the transducer which was subsequently converted to a voltage and used to trigger the MR image acquisition. In addition to serving as a trigger for the MR acquisition, the pressure transducer voltage output was recorded at a sampling rate of 200 Hz (averaged during analysis to produce curves of mean force) and later converted into units of force (N) based on a calibration of the system using disc weights. The Maximum Voluntary Contraction (MVC) was determined for each subject as the best of three trials recorded prior to MR imaging. MVC was measured for each ankle position and sub-maximal contraction levels were set based on the MVC at that ankle position.

Images were acquired under different experimental conditions: during two submaximal, isometric contraction of the plantarflexor muscles (at 25% and 50% of the subject's MVC) with the foot at three ankle positions: dorsiflexed (DF), neutral (N), plantarflexed (PF). MR image acquisition required ~53 repeated contractions; thus, it was important to ensure consistency of motion. The subject was provided with the feedback of the actual force generated by the subject superposed on the desired force curve to facilitate consistent contractions.

Imaging Protocol: A large FOV sagittal scout was acquired using the body coil in order to measure the ankle angle from the images. A set of high-resolution water-saturated oblique sagittal fast spin echo (FSE) images of the MG (TE: 12.9 ms, TR: 925 ms, NEX: 4, slice thickness: 3 mm, interslice gap: 0 mm, FOV: 30 cm × 22.5 cm, 512 × 384 matrix) was initially acquired. This sequence provides high-tissue contrast from the high signal from fat in fascicles in the background of suppressed muscle water signal and was used to visualize fascicles. The slice that best depicted the fascicles was selected for the Velocity-Encoded Phase-Contrast (VE-PC) scan. The VE-PC sequence had three-directional velocity encoding and a single oblique sagittal slice was acquired (TE: 7.7 ms, TR: 16.4 ms, NEX: 2, FA: 20°, slice thickness: 5 mm, FOV: 30 cm × 22.5 cm, partial-phase FOV = 0.55, 256 × 192 matrix, 4 views/segment, 1 slice, 22 phases, 10 cm·s$^{-1}$ 3 direction velocity encoding). This resulted in 53 repetitions [([192 (phase encode lines) × 0.55 (partial FOV) × 2 (NEX)/4 (views per segment) = 53])] for the image acquisition. In total, twenty-two phases (using view-sharing) were collected within each contraction–relaxation cycle of ~3 s (isometric contraction). At each ankle position, diffusion tensor images (DTI) using 30 diffusion directions at b = 400 s/mm$^2$ with geometric parameters matched to the VE-PC images were acquired. The study protocol included at each ankle angle: large FOV image, VE-PC at 50%, followed by the 5-min DTI acquisition and then 25%MVC. The foot was then repositioned before repeating the imaging protocol. It should be noted that this order of imaging was implemented to minimize fatigue between the different dynamic acquisitions. The DTI data are not used in the current study but in a separate analysis to identify fascicles to derive fiber strains.

Image Analysis: The phase images of the VE-PC data directly quantify velocity in the direction of the velocity-encoding gradient. Prior to extracting the velocity data, the phase images were corrected for phase artifacts arising from sources such as B$_0$ inhomogeneities and chemical shift and not from the velocity-encoding gradient. Velocity images extracted from the phase-corrected data are inherently noisy. As the calculation of the strain or SR tensor involves estimation of the spatial gradients of the displacement/velocity images that introduces additional noise into the image, the velocity images were first denoised using a 2D anisotropic diffusion filter [16]. The anisotropic diffusion filter reduces noise in homogenous regions while preserving edges, maintaining the effective resolution of the original velocity image. The filter was applied iteratively to reduce noise in homogenous regions, and was defined by the equation:

$$c(||\nabla I||) = \exp\left(-\left(\frac{||\nabla I||}{K}\right)^2\right) \quad (1)$$

where $c$ is the diffusion coefficient, $I$ is the image to be denoised, and $\nabla I$ is the image gradient. The extent of denoising is controlled by the value of $K$ and the number of iterations. $K$ was held at a low value of 2 since there are no strong edges in the phase images. The level of denoising was explored at two values of the number of iterations, N: 10 and 15. The number of iterations at 10 was chosen as an optimum, a trade-off between noise reduction in the strain indices and excessive blurring. Figure S1 shows the phase images acquired from velocity encoded in the *x*-, *y*- and *z*-directions, respectively, along with corresponding noise-filtered images at two values of N (10 and 15). The reduction in noise is readily visualized at both iterations while an increase in blurring is seen at N = 15 iteration.

Strain and SR tensor: Voxels in the entire volume were tracked to obtain (in-plane) displacements using the velocity information in the phase images. The displacement maps (in *x*- and *y*-directions) as well as the velocity maps were processed as outlined below to obtain 2D strain and strain rate tensors in the principal frame of reference. For each voxel in the displacement and velocity maps, the 2D spatial gradient maps, *L*, of the displacement and velocity vector were calculated as detailed in [1]:

$$L = \begin{bmatrix} \frac{\partial u}{\partial x} & \frac{\partial v}{\partial x} \\ \frac{\partial u}{\partial y} & \frac{\partial v}{\partial y} \end{bmatrix} \tag{2}$$

where *u* and *v* are the *x* and *y* components of either the displacement or the velocity vector. The symmetric form of the spatial gradient of displacement or velocity is generated from:

$$D = 0.5 \left( L + L^{\mathrm{T}} \right) \tag{3}$$

The symmetric tensor *D* was diagonalized to yield the 2 × 2 strain (*E*, Eulerian strain) and *SR* tensor in the principal frame of reference. The principal components of the strain or strain rate tensor (eigenvalues arranged in ascending order) were labeled as follows: $E_{\lambda 1}$ and $E_{\lambda 2}$ are the normal principal strains while $SR_{\lambda 1}$ and $SR_{\lambda 2}$, the normal principal strain rates (defined as perpendicular to the face of an element and represented by the diagonal terms of the *E* or *SR* tensor). It should be noted that during the compression part of the cycle, the eigenvector corresponding to $E_{\lambda 1}$ and $SR_{\lambda 1}$ is in a direction close to the muscle fiber direction while in the relaxation phase the eigenvectors are in a direction approximately orthogonal to the fiber direction. The reverse is true for $E_{\lambda 2}$ and $SR_{\lambda 2}$. Two other strain and *SR* indices are calculated from the tensors: the out-of-plane strain denoted by $E_{\text{out-plane}}$ and $SR_{\text{out-plane}}$ and the maximum shear strain or shear strain rate denoted by $E_{\text{max}}$ and $SR_{\text{max}}$, respectively. The out-of-plane strain and strain rate, which is in the fiber cross-section perpendicular to the imaging plane, was calculated from the sum of the principal eigenvalues at each voxel based on the assumption that muscle tissue is incompressible. A local contraction along the muscle will be accompanied by a local expansion in the plane perpendicular to the fiber. If considered in 3D, the sum of the three strain rates for an incompressible volume should be zero. However, only the 2D tensor is calculated here due to the constraints of single slice imaging, so the negative of the sum of the two eigenvalues (*E* or *SR*) yields the magnitude of the third eigenvalue.

$$E_{\text{out-plane}} = -\left( E_{\lambda_1} + E_{\lambda_2} \right) \tag{4a}$$

$$SR_{\text{out-plane}} = -\left( SR_{\lambda_1} + SR_{\lambda_2} \right) \tag{4b}$$

Shear strain and strain rate (represented by the off-diagonal terms of the *E* and *SR* tensors) are dependent on the frame of reference; it is zero in the principal frame and is a maximum when the 2D tensor is rotated from the principal frame by 45°. In this frame, the diagonal terms are zero and one can obtain the maximum shear strain or strain rate. Mathematically, the maximum shear strain or strain rate is also found from:

$$E_{\text{max}} = 0.5 \left( E_{\lambda_1} - E_{\lambda_2} \right) \tag{5a}$$

$$SR_{\text{max}} = 0.5 \left( SR_{\lambda_1} - SR_{\lambda_2} \right) \tag{5b}$$

Shear strains or strain rates are defined as parallel to the face of an element and represented by off-diagonal terms in the *E* or *SR* tensor).

Strain and SR indices in ROIs positioned at the proximal, middle and distal regions of the MG muscle (corresponding to a location at approximately 75%, 50% and 25% of the total length of the MG) were extracted. The ROIs were positioned in the first frame of the

dynamic data and the pixels in the ROI were tracked using the velocity data to ensure that the measurement is performed on the same pixels even if they have moved to different locations. The tracked ROIs were also checked manually in a cine loop to confirm that the entire ROI stayed in the MG. Statistical analysis was performed on the average of values extracted from the proximal and middle ROIs; the distal ROI was not used as the values were noisy. An analysis of the spatial variation of the strain and strain rate between the proximal and middle regions would be interesting. However, the current study lacks the statistical power to introduce another factor in the analysis; this will be the subject of a future study. Values of all strain and strain rate indices values were extracted from the temporal frame corresponding to the peak in $SR_{\lambda 1}$ for all strain rate indices and at the peak of $E_{\lambda 1}$ for the strain indices.

Strain Deformation in the Fiber Cross-Section: In the following discussion, all strain values refer to absolute values.

Case 1: Symmetric deformation in fiber cross-section: In this case, the deformation in the fiber cross-section is symmetric, i.e., the deformation is the same in both directions. This leads to $E_{\lambda 2} \sim E_{\text{out-plane}}$ and by the incompressibility of muscle tissue, these two strains will be equal to the half of $E_{\lambda 1}$. The maximum shear strain will be $\sim 0.75 |E_{\lambda 1}|$. This is illustrated in Figure 2a.

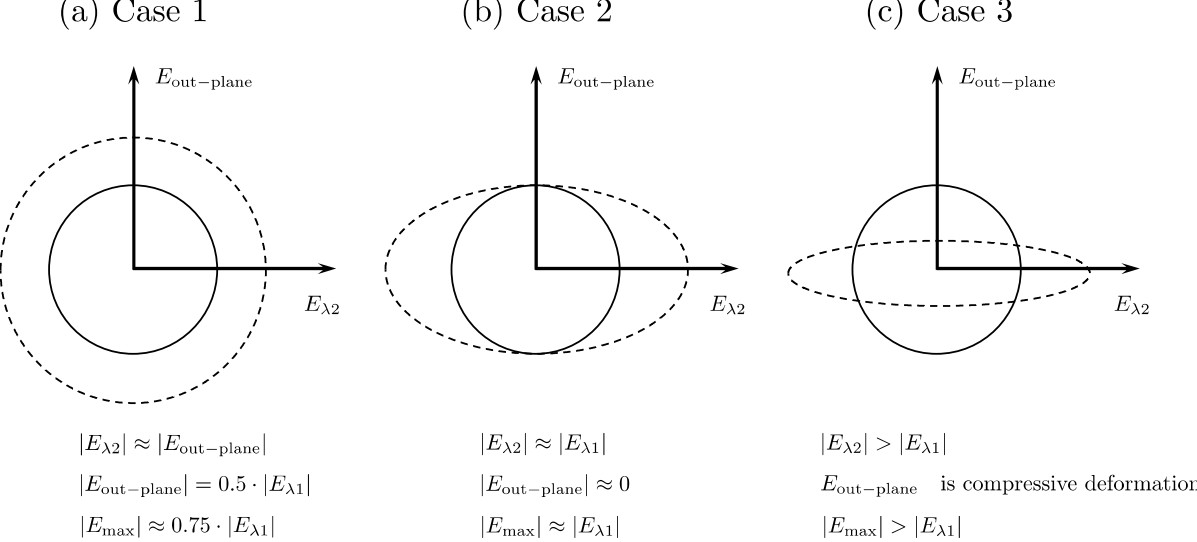

**Figure 2.** Schematic of different deformation patterns in the fiber cross-section, the non-deformed state is shown as a solid line and the deformed state with a dashed line. (**a**): Symmetric deformation in fiber cross-section leads to $|E_{\lambda 2}| \sim |E_{\text{out-plane}}| \sim 0.5 |E_{\lambda 1}|$ and $|E_{\text{max}}| \sim 0.75 |E_{\lambda 1}|$. (**b**): Asymmetric deformation in fiber cross-section with little to no deformation in the out-plane direction leads to $|E_{\lambda 2}| \sim |E_{\lambda 1}|$, $E_{\text{out-plane}} \sim 0$ and $|E_{\text{max}}| \sim |E_{\lambda 1}|$. (**c**): Highly asymmetric deformation in fiber cross-section with $|E_{\lambda 2}| > |E_{\lambda 1}|$ will lead to $E_{\text{out-plane}}$ being negative (compressive strain in the fiber cross-section) and $|E_{\text{max}}| > |E_{\lambda 1}|$.

Case 2: Asymmetric deformation in fiber cross-section: In this case, the deformation in the fiber cross-section is along one direction, say along $E_{\lambda 2}$. Then, deriving from the incompressibility of muscle tissue, $E_{\text{out-plane}}$ will be close to zero or very low values and $E_{\lambda 2}$ will be equal to $E_{\lambda 1}$. The maximum shear strain will be $\sim E_{\lambda 1}$. This is illustrated in Figure 2b.

Case 3: Highly asymmetric deformation in the fiber cross-section with $E_{\lambda 2}$ greater than $E_{\lambda 1}$. In this case, the deformation in the fiber cross-section is such that the radial expansion in the in-plane direction exceeds that of the compressive strain in the fiber direction. From the incompressibility of the muscle tissue, this will lead to a compressive deformation in the out-plane direction. The maximum shear strain will be greater than $E_{\lambda 1}$. This is illustrated in Figure 2c.

Statistical analysis: The outcome variables of the analysis are the eigenvalues of the strain tensor ($E_{\lambda 1}$, $E_{\lambda 2}$, $E_{\text{out-plane}}$, $E_{\text{max}}$) and the strain rate tensor ($SR_{\lambda 1}$, $SR_{\lambda 2}$, $SR_{\text{out-plane}}$, $SR_{\text{max}}$). Strain is unitless and the SR eigenvalues are in units of $\text{s}^{-1}$. Normality of data was tested by using both, the Shapiro–Wilke test and visual inspection of Q-Q plots. Principal strains and strains rates as well the normalized strains and strain rates were normally distributed. Thus, changes between ankle angles, %MVC as well as potential interaction effects (ankle angle $\times$ %MVC), were assessed using two-way repeated measures ANOVAs and in case of significant ANOVA results for the factor 'ankle angles', Bonferroni-adjusted post hoc analyses were performed. Data are reported as mean $\pm$ SD for the variables since they were normally distributed. For all tests, the level of significance was set at $\alpha = 0.05$. In addition to the above statistical tests, exploratory analysis using paired *t*-test was performed at each ankle angle between (i) absolute values of $E_{\lambda 1}$ and $E_{\lambda 2}$ using data from both force levels and (ii) absolute values of $E_{\lambda 1}$ and $E_{\text{max}}$ using data from both force levels. The statistical analyses were carried out using SPSS for Mac OSX (SPSS 28.0.1.1, SPSS Inc., Chicago, IL, USA).

### 3. Results

*3.1. MVC at Different Ankle Angles*

Maximum Voluntary Contraction (MVC) was measured for each subject at each ankle angle as the best of three trials recorded prior to imaging: $MVC_{\text{DF}} = 289 \pm 9$ N, $MVC_{\text{N}} = 143 \pm 14$ N, $MVC_{\text{PF}} = 65 \pm 10$ N (average over all 6 subjects). The MVCs were significantly different between the three ankle angles: $MVC_{\text{DF-N}}$ ($p = 0.0012$), $MVC_{\text{N-PF}}$ ($p = 0.0003$), and $MVC_{\text{DF-PF}}$ ($p = 0.0012$), where the subscripts are the two ankle angles compared in paired *t*-tests.

*3.2. Strain and Strain Rate Maps and Temporal Plots of Deformation Indices*

Figure 3 shows for one subject, the compressive strain ($E_{\lambda 1}$) and strain rate ($SR_{\lambda 1}$) maps through select frames of the dynamic cycle of 22 temporal frames for the three ankle angles at 50%MVC. The values of $E_{\lambda 1}$ and $SR_{\lambda 1}$ are superposed on the magnitude images of the VE-PC dataset using a colormap. Negative values of strain or strain rate (blue hue) are seen in the medial gastrocnemius and in the soleus (plantar flexor muscles) around frame 11 for the strain and around frames 8 and 16 for the strain rate maps. The temporal variation of strain and strain rate are shown in Figure 4a,b for one subject for a ROI placed in middle of the MG muscle for the three ankle angles and two %MVCs. Figure S2 shows for one subject, $E_{\lambda 2}$ and $SR_{\lambda 2}$ maps through select frames of the dynamic cycle of 22 temporal frames for the three ankle angles at 50%MVC. The values of $E_{\lambda 2}$ and $SR_{\lambda 2}$ are superposed on the magnitude images of the VE-PC dataset using a colormap. Positive values of strain or strain rate (red hue) are seen in the medial gastrocnemius and in the soleus (plantar flexor muscles) around frame 11 for the strain and around frames 8 and 16 for the strain rate maps.

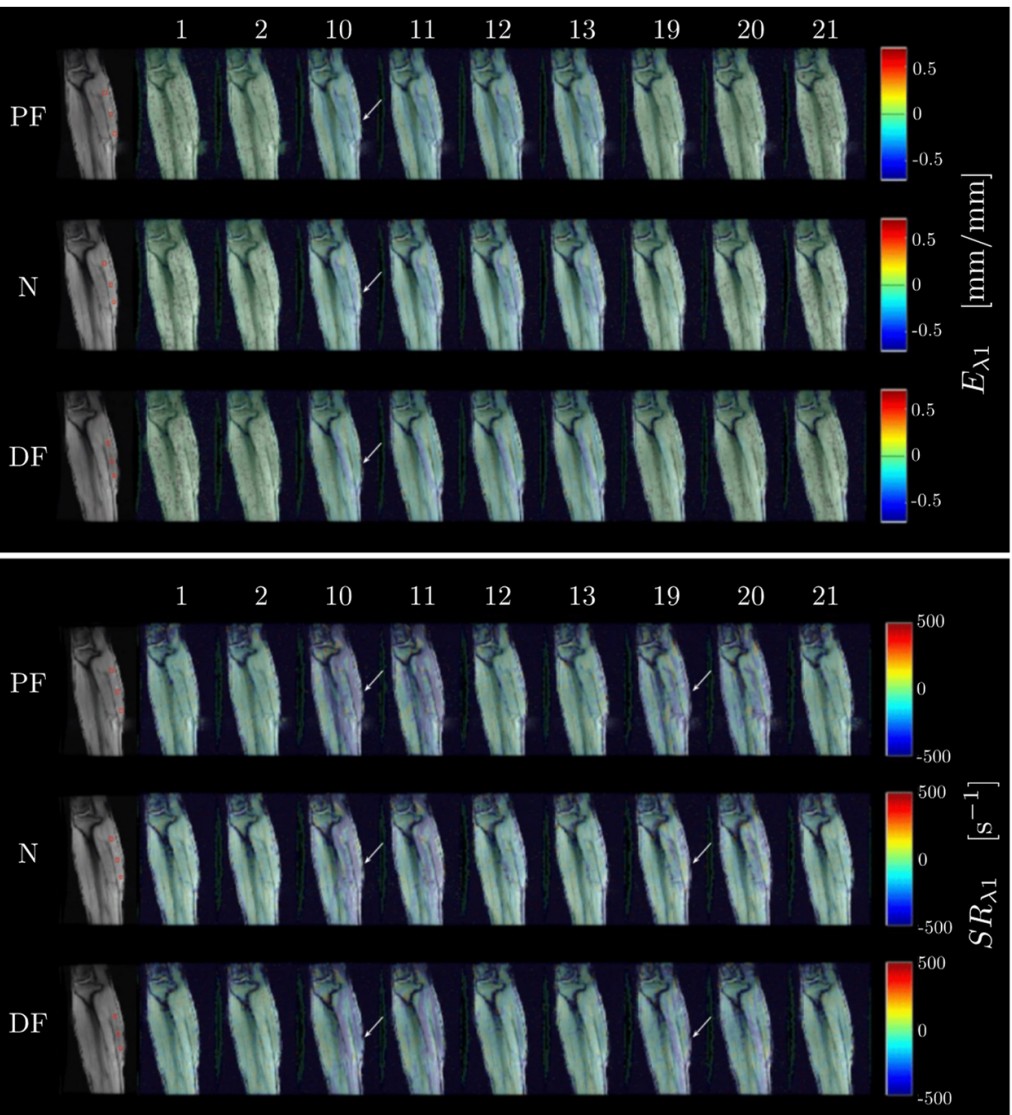

**Figure 3.** Maps of the negative strain ($E_{\lambda 1}$) (**top** panel) and negative strain rate ($SR_{\lambda 1}$) (**bottom** panel) projected on magnitude images at the corresponding temporal frame. Images shown here were acquired at 50%MVC, at each foot position PF, N, DF (order of rows **top** to **bottom**). Overlay allows for better identification of the underlying muscle, aponeuroses, and fascicles. The color maps are color-coded according to the legend with the figures. Select frames (from the acquired 22 dynamic frames) where peak strains and strain rates occur are shown here, the frame number is indicated on the top row. The peak of the strain occurs around frame 10–11 (arrow points to MG) where the blue shade corresponding to compressive strains in the MG and in the soleus can be seen. The peak of the strain rate occurs in the contraction (~frame 7–8, arrow to MG) and relaxation (~frame 16) phases and this is visualized as blue shades in the MG and in the soleus around these frames. The regions of interest in the MG (seen as red boxes in the first frame of each row) used to extract the deformation indices are shown in the first frame.

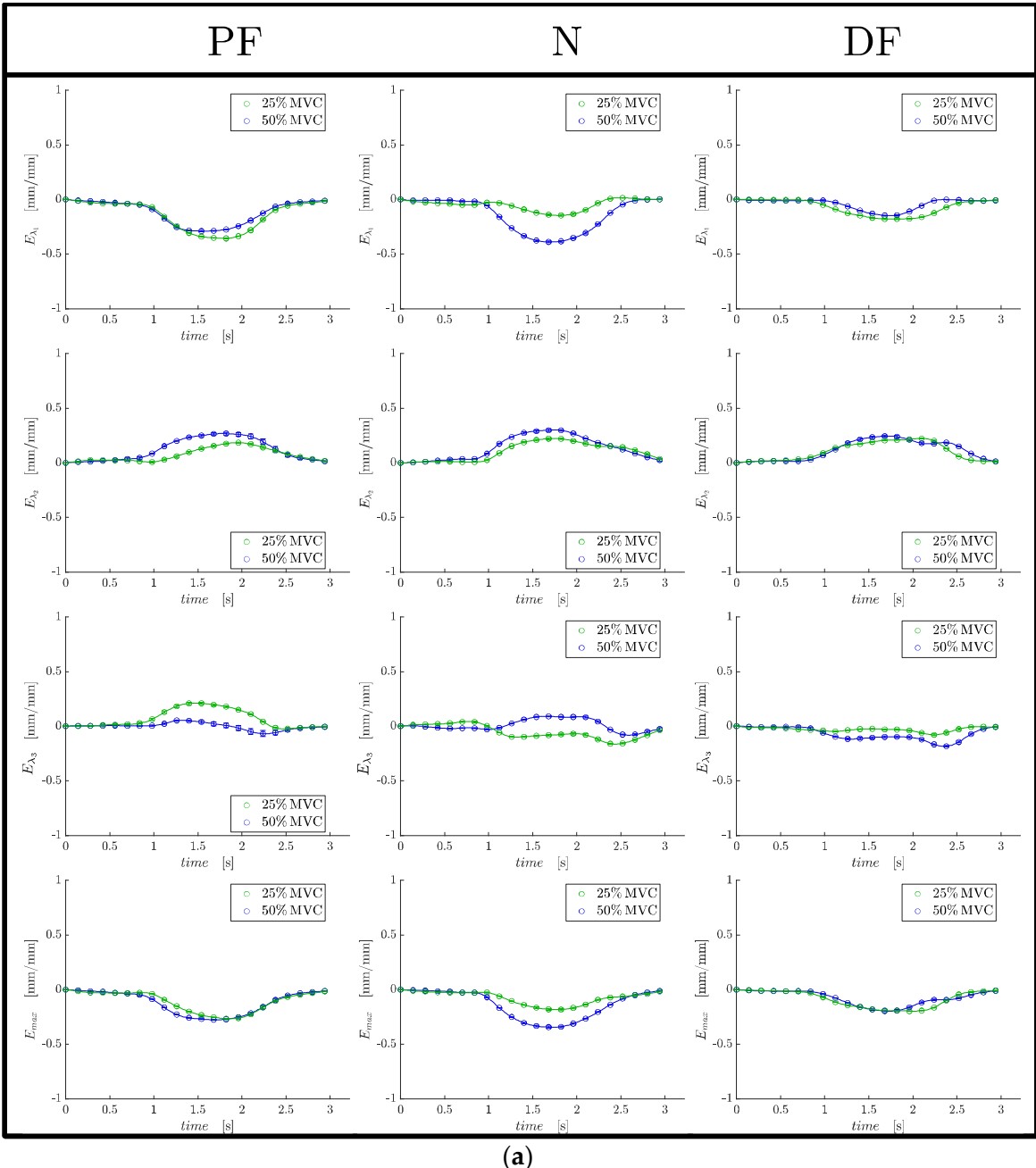

(**a**)

**Figure 4.** *Cont.*

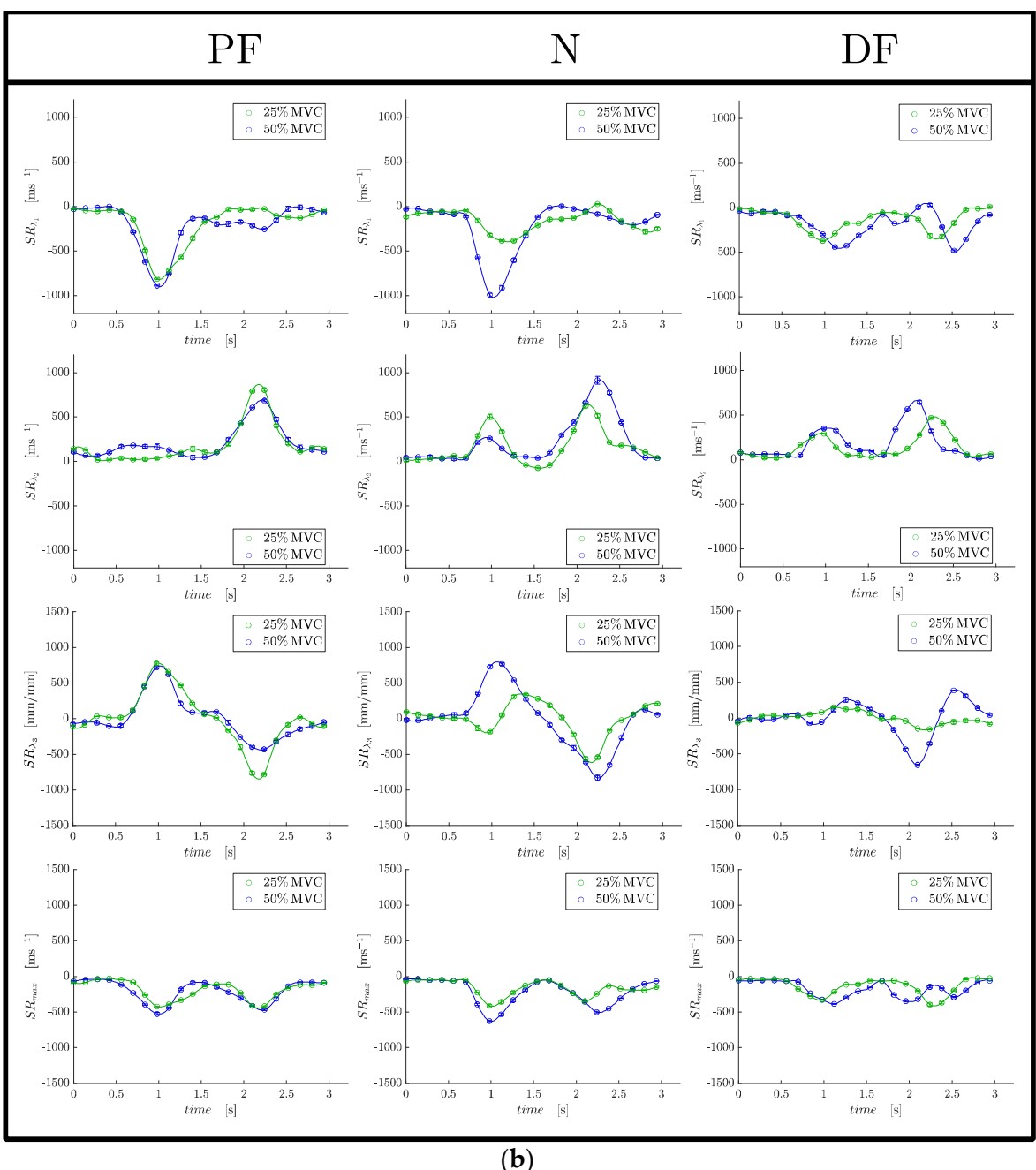

**Figure 4.** (**a**) Temporal plots for strain indices for one subject for an ROI placed in the middle of the MG muscle for the three ankle angles (PF, N, DF). Organized in column by foot angle position, and in row order of: $E_{\lambda 1}$, $E_{\lambda 2}$, $E_{\lambda 3 \text{ (or out-plane)}}$, and $E_{\max}$. (**b**) Temporal plots for strain rate indices for one subject for an ROI placed in the middle of the MG muscle for the three ankle angles (PF, N, DF). Organized in column by foot angle position, and in row order of: $SR_{\lambda 1}$, $SR_{\lambda 2}$, $SR_{\lambda 3 \text{ (or out-plane)}}$, and $SR_{\max}$.

### 3.3. Region of Interest Values of Deformation Indices at Peak Contraction

Table 1 and Table S1 list the peak strain and strain rate indices, respectively, averaged over all subjects for the three ankle angles and %MVC (data shown is the average for proximal and middle ROIs). Compressive strain and strain rate values ($E_{\lambda 1}$ and $SR_{\lambda 1}$) showed significantly lower absolute values at 25%MVC compared to 50%MVC while showing no significant changes between ankle angles. Radial strains and strain rates ($E_{\lambda 2}$, $SR_{\lambda 2}$, $E_{\text{out-plane}}$, $SR_{\text{out-plane}}$) in the fiber cross-section showed no significant changes with

%MVC or with ankle angle though $E_{\lambda 2}$ was consistently higher at 50%MVC. The absolute values of $E_{\text{out-plane}}$ and $SR_{\text{out-plane}}$ are much smaller than the strains and strain rates in the other two directions, indicating that deformation is the smallest in this direction. Further, while $E_{\lambda 2}$ and $SR_{\lambda 2}$ are clearly radial expansion strain and strain rates, respectively, in the fiber cross-section, $E_{\text{out-plane}}$ and $SR_{\text{out-plane}}$ are smaller in magnitude and exhibit negative signs indicating that these are small radial compressive strain and strain rates, respectively, in the out-of-plane direction. This indicates that the deformation is close to that shown in Figure 2, between Case 2 and Case 3. It should be noted that the largest absolute values of $E_{\text{out-plane}}$ and $SR_{\text{out-plane}}$ occur for the dorsiflexed ankle position. The maximum shear strain showed no significant changes with either %MVC (though larger absolute values at 50%MVC) or ankle angle. Maximum shear strain rate showed significant changes with %MVC (higher absolute values at 50%MVC) but no significant change with ankle angle.

**Table 1.** Strain indices for different ankle angles and %MVC.

| Ankle Position | %MVC | Peak Force (N) | $E_{\lambda 1}$ * | $E_{\lambda 2}$ | $E_{\text{out-plane}}$ | $E_{\text{max}}$ |
|---|---|---|---|---|---|---|
| Plantarflexion | 50 | $32 \pm 6.959$ | $-0.141 \pm 0.009$ | $0.168 \pm 0.03$ | $-0.036 \pm 0.031$ | $-0.151 \pm 0.019$ |
| | 25 | $16.72 \pm 3.694$ | $-0.102 \pm 0.022$ | $0.137 \pm 0.031$ | $-0.046 \pm 0.019$ | $-0.117 \pm 0.026$ |
| Neutral | 50 | $74.24 \pm 9.655$ | $-0.185 \pm 0.015$ | $0.208 \pm 0.025$ | $-0.013 \pm 0.033$ | $-0.19 \pm 0.014$ |
| | 25 | $37.68 \pm 4.883$ | $-0.134 \pm 0.024$ | $0.202 \pm 0.037$ | $-0.068 \pm 0.052$ | $-0.165 \pm 0.022$ |
| Dorsiflexion | 50 | $141.2 \pm 9.246$ | $-0.143 \pm 0.019$ | $0.209 \pm 0.008$ | $-0.083 \pm 0.023$ | $-0.173 \pm 0.006$ |
| | 25 | $73.82 \pm 4.26$ | $-0.095 \pm 0.027$ | $0.169 \pm 0.018$ | $-0.092 \pm 0.025$ | $-0.129 \pm 0.019$ |

* Significant difference between 25% and 50%MVC.

Table 2 and Table S2 list the peak strain indices normalized to force and peak strain rate indices normalized to force, respectively, averaged over all subjects for the three ankle angles and %MVC (data shown are the average for proximal and middle ROIs). Comparing the absolute values across all %MVC and ankle angles, the lowest normalized strains were at 50%MVC and the dorsiflexed ankle angle while the highest values were at the 25%MVC and the plantarflexed ankle angle. $E_{\lambda 1}$ normalized to force significantly changed with ankle angle, pairwise comparison revealed changes between (PF and DF), (PF and N), (N and DF) but showed no significant change with %MVC. $SR_{\lambda 1}$ normalized to force significantly changed with %MVC and also with ankle angle, pairwise comparison revealed changes between PF and N. $E_{\lambda 2}$ normalized to force showed significant change with %MVC as well as with ankle angles, pairwise comparison revealed changes between (PF and DF) and (N and DF). $SR_{\lambda 2}$ normalized to force showed significant change with %MVC as well as with ankle angles, pairwise comparison revealed changes between (PF and N) and (PF and DF). Normalized $E_{\text{out-plane}}$ and $SR_{\text{out-plane}}$ showed no significant changes with either %MVC or with ankle angle. $E_{\text{max}}$ and $SR_{\text{max}}$ normalized to force significantly changed with %MVC and also with ankle angle, pairwise comparison revealed changes between (PF and DF), (N and DF).

Table 3 lists the absolute value of $E_{\lambda 1}$, $E_{\lambda 2}$ and $E_{\text{max}}$ for each ankle angle averaged over all subjects and both force levels. The mean of $E_{\lambda 2}$ was greater than $E_{\lambda 1}$ for all three ankle angles and paired *t*-tests comparing the absolute values of $E_{\lambda 1}$ and $E_{\lambda 2}$ yielded the following results: for PF ankle angle, the absolute values of $E_{\lambda 1}$ and $E_{\lambda 2}$ were significantly different ($p = 0.04$), for neutral ankle angle the difference was not significant while for the dorsiflexed ankle the difference was highly significant ($p = 0.002$) and the largest difference in means of the absolute values of $E_{\lambda 1}$ and $E_{\lambda 2}$ was seen in the DF ankle angle (~59% compared to ~26% for the other two ankle angles). The mean of $E_{\text{max}}$ was greater than $E_{\lambda 1}$ for all three ankle angles and paired *t*-tests comparing the absolute values of $E_{\lambda 1}$ and $E_{\text{max}}$ showed significant differences between the two values only for the dorsiflexed ankle angle ($p = 0.004$) and the size effect was also largest at the DF ankle angle (~27% compared to ~10%).

**Table 2.** Strain indices normalized to force for different ankle angles and %MVC.

| Ankle Position | %MVC | Peak Force (N) | $E_{\lambda 1}$ [o,!,δ] | $E_{\lambda 2}$ [o,!,*] | $E_{\text{out-plane}}$ | $E_{\text{max}}$ [o,!,*] |
|---|---|---|---|---|---|---|
| PF | 50 | $32 \pm 6.96$ | $-0.0053 \pm 0.001$ | $0.0061 \pm 0.0012$ | $-0.001 \pm 0.0016$ | $-0.0056 \pm 0.001$ |
|    | 25 | $16.72 \pm 3.69$ | $-0.0069 \pm 0.0007$ | $0.01 \pm 0.0012$ | $-0.0041 \pm 0.0009$ | $-0.0083 \pm 0.0009$ |
| N | 50 | $74.24 \pm 9.65$ | $-0.0027 \pm 0.0004$ | $0.0032 \pm 0.0004$ | $-0.0003 \pm 0.0004$ | $-0.0028 \pm 0.0003$ |
|   | 25 | $37.68 \pm 4.88$ | $-0.0043 \pm 0.0011$ | $0.0058 \pm 0.0007$ | $-0.0012 \pm 0.0018$ | $-0.005 \pm 0.0006$ |
| DF | 50 | $141.2 \pm 9.25$ | $-0.001 \pm 0.0002$ | $0.0015 \pm 0.0001$ | $-0.0006 \pm 0.0002$ | $-0.0012 \pm 0.0001$ |
|    | 25 | $73.82 \pm 4.26$ | $-0.0015 \pm 0.0005$ | $0.0025 \pm 0.0003$ | $-0.0013 \pm 0.0004$ | $-0.0019 \pm 0.0003$ |

* Significant difference between 25% and 50%MVC; ! Significant difference between PF and DF; δ Significant difference between PF and N; o Significant difference between N and DF.

**Table 3.** Comparison of absolute values (std dev) of $E_{\lambda 1}$ to $E_{\lambda 2}$ and $E_{\lambda 1}$ to $E_{\text{max}}$ for PF, N, DF.

| Ankle Position | Abs ($E_{\lambda 1}$) | $E_{\lambda 2}$ | Abs ($E_{\text{max}}$) | %Diff ($E_{\lambda 1}$, $E_{\lambda 2}$) | %Diff ($E_{\lambda 1}$, $E_{\text{max}}$) |
|---|---|---|---|---|---|
| Plantarflexion * | 0.121(0.044) | 0.152(0.073) | 0.134(0.056) | 25.50% | 10.10% |
| Neutral | 0.160(0.054) | 0.205(0.073) | 0.178(0.044) | 28.60% | 11.40% |
| Dorsiflexion *,** | 0.119(0.060) | 0.189(0.039) | 0.151(0.040) | 59.40% | 27.30% |

* Significant difference between absolute values of $E_{\lambda 1}$ and $E_{\lambda 2}$; ** Significant difference between $E_{\lambda 1}$ and $E_{\text{max}}$.

## 4. Discussion

The force attained at Maximum Voluntary Contraction (MVC) at the three ankle angles were significantly different ($p < 0.001$) with the maximum MVC at the dorsiflexed ankle position and the minimum MVC at the plantarflexed position. This clearly shows that the most efficient ankle position for force generation is the dorsiflexed position. This has also been observed in previous studies [13–15]. Moreover, this finding highlights the importance of maintaining the ankle angle in the same position for longitudinal or cross-sectional dynamic cohort studies (e.g., young vs. old subjects).

Strain and strain rate are deformation indices; strain requires a frame of reference while strain rate is an instantaneous measurement. Strain rate is equal to differential velocities of the tissue, while strain is equal to differential displacement. Thus, strain rate maps reported herein are derived from the acquired velocity images, while the strain maps are computed from displacement maps tracked from the acquired velocity images. While strain and strain rate are related, SR can be different for tissue regions having the same strain. This is the first study of principal strains and strain rates during isometric contraction at different ankle positions. It has the advantage that compared to fiber strains, there is no need to identify the direction of the muscle fibers. In ultrasound and prior dynamic MR studies, muscle fibers were identified via fascicle locations [1,13]. This latter process is prone to error (since fascicles may not entirely run in the plane of the image) and suffers from low contrast (e.g., the lack of fat tissue in young subjects close to the fascicles prevents visualization of the fascicles in MRI). Further, the 2D strain/SR tensor analysis used in the current paper provides measurements of the deformation in the fiber cross-section as well as the shear strain in addition to the contractile strain. In contrast, 1D strain analysis (either US or MRI) can only provide information about muscle contractility [13]. In-plane deformation and shear strain/SR are influenced by the material properties of the extracellular matrix (ECM) and thus, measurement of the in-plane deformation and shear strain/shear strain rate with ankle angle could potentially provide information on the effect of the ECM on force production. Further, the ability to measure or deduce deformations in the fiber cross-section affords an opportunity to monitor fiber shape changes during isometric contraction at the different ankle angle positions.

At each ankle position, absolute values of the strain indices increased with submaximal %MVC, although it was significant only for $E_{\lambda 1}$, $SR_{\lambda 1}$ and $SR_{\text{max}}$ (Table 1). Significant differences in $E_{\lambda 1}$ and $SR_{\lambda 1}$ with %MVC are anticipated as the higher force at the higher %MVC requires a larger contraction (strain). Surprisingly, there were no significant differences in the strain or SR indices between the different ankle angles despite a highly

significant difference in force between the different ankle angles. This implies that similar strains (amount of contraction) at the different ankle angles were capable of producing significantly different forces. The deduced absolute values of $E_{out-plane}$ and $SR_{out-plane}$ are much smaller than the strains and strain rates in the orthogonal direction of the fiber cross-section indicating a strong anisotropy of deformation; this is true for all ankle angles. Anisotropy of fiber cross-section deformation has been reported in earlier studies for the neutral ankle angle [1–4,17]. The results from this study also show that anisotropy holds at plantarflexed and dorsiflexed ankle angles.

The normalized strain indices (normalized to the force for the ankle position/%MVC) showed significant differences for both force levels and ankle positions (Table 2). The absolute value of the normalized strain/SR indices was higher for the lower %MVC than for the higher %MVC: this implies a lack of linearity between strain and %MVC and that larger contraction/force (strains) is needed to achieve lower %MVCs. This may also arise from the MG contributing more to force production at lower %MVCs and the soleus contributing more at the higher %MVCs. All the normalized strain and strain rate indices were lower in the dorsiflexed position than in the plantarflexed or neutral ankle positions and most of these changes were significant (Table 2). The changes in the normalized principal compressive strain can be understood in terms of the muscle force–length (FL) curve. The FL relationship describes the dependence of the steady-state isometric force of a muscle fiber or sarcomere on muscle fiber or sarcomere length and the 'sliding filament' theory has been used to explain the FL curve [10,11]. In this theory, the maximal isometric force of a sarcomere is determined by the amount of overlap between the contractile filaments, actin, and myosin [10]. Starting from short muscle lengths, force increases as sarcomere length increases (ascending slope), reaches a plateau at intermediate lengths (optimal length for maximum force production), followed by a decrease in force as sarcomere length increases (descending slope) at long muscle lengths. Lower normalized strain at the DF ankle position implies that small contractions at this ankle angle are capable of producing large forces from which it can be deduced that the muscle fiber length in the dorsiflexed position is close to the optimal length of the force–length curve. Compared to the dorsiflexed position, the plantarflexed position is inefficient for force production—essentially implying that it is far from the optimal length for isometric force production. Normalized strains in the neutral ankle position were intermediate between the plantar and dorsiflexion states. Prior studies have identified that, in vivo, the plantar flexors work on the ascending limb of the force–length relationship due to the anatomical constraints of the ankle- and knee-joints [11–13]. The lower force of the plantarflexed angle can thus be attributed to the shorter length of the muscle fiber at this ankle angle which places it lower on the ascending limb of the FL curve and consequently, lower force production.

Arampatzis et al. studied MG fiber lengths and electrical activity using US and EMG, respectively [13]. They reported that active fiber lengths at MVC were not significantly different between knee flexion/ankle angle positions even though resting fiber lengths were significantly different between knee flexion/ankle angle positions [13]. Furthermore, EMG was significantly reduced in the most plantarflexed position despite active fiber lengths being the same in all the knee flexion/ankle angle positions. Arampatzis et al. identified that the main mechanism for the decrease in EMG activity is a neural inhibition mechanism [13]. This neural inhibition occurs because the muscle reaches a critical shortened length and since it is further down in the ascending limb of the force–length relationship, the torque output cannot be increased even if the muscle is fully activated [18]. Compared to Arampatzis' study that reported a reduction in EMG at the plantarflexed position, the current study did not see a reduction in strain (no significant difference in strain or SR between the ankle angles) at PF [13]. One source of discrepancy could be that the maximum force level was at 50%MVC in the current study compared to 100%MVC for the US/EMG studies. However, it should be noted that the current study, similar to previous studies, also identified the plantarflexed position as the least efficient in force production.

The above analysis of the contractile strain/SR attributes the lower force generation at the plantarflexed ankle angle (compared to the neutral/dorsiflexed ankle angle) entirely to the critical shortened length and the relative position on the force length curve for the three ankle angles. While this is likely the biggest contributor to the reduced force and to the increase in normalized strains at PF (larger strains/force compared to the DF ankle angle), the current analysis shows there may potentially be other contributors to the loss of force. Azizi et al. advanced the hypothesis that constraints to radial expansion in the fiber cross-section could limit the extent of contraction, limiting force generation and verified this in a physical model and in vivo; the latter by applying external constraints in the muscle cross-section [19]. An analysis of the absolute values of $E_{\lambda 1}$ and $E_{\lambda 2}$ for the three ankles showed that the dorsiflexed ankle position had the largest and most significant difference ($|E_{\lambda 2}| > |E_{\lambda 1}|$); the deformation is similar to that shown in Figure 2c. On the other hand, while $|E_{\lambda 2}| > |E_{\lambda 1}|$ for both plantarflexor and neutral ankle angles, the differences were smaller and tentatively, the deformation patterns in these two ankle angles may be between the schematics shown in Figure 2b,c. One explanation for the PF and N to have smaller in-plane deformations (smaller $E_{\lambda 2}$) compared to the dorsiflexed position may be related to the initial (at rest) fiber radial size. The plantarflexed position has the largest fiber cross-section of the three ankle angles and the larger initial radial size may provide a constraint to further radial expansion. Azizi et. al. showed with a physical model that constraints to radial expansion limits the contractility and thus, the force generated [19]. Thus, the constraints to radial expansion in the PF (arising from the larger radius) may also be a contributor to force reduction in this ankle angle position. Further, computational modeling studies have predicted that when there is a strongly anisotropic constraint the force output may increase by a factor of two [8]. This latter computational model showed that maximum force output was obtained by introducing anisotropy of passive material stiffness along the fiber cross-sectional axes such that there was very little deformation along one axis (the through-plane axis) during a muscle length change. In this anisotropic model, the stiffness in one direction was reinforced such that it was stiffer by a factor of 4 compared to the orthogonal direction that resulted in a near doubling in force compared to an isotropically stiff material. The authors postulated that the structural muscle proteins called costameres were a potential candidate for introducing such an anisotropy in the passive material properties [8]. Highly asymmetric deformation in the fiber cross-section seen in DF may be facilitated since in this ankle position, the fiber is longest and consequently, the fiber cross-section area is the smallest allowing larger radial expansions. A strongly anisotropic constraint, as is seen in DF, provides another potential mechanism of higher force in DF from the highly asymmetric deformation at this ankle angle. It should be noted that the strain in the fiber cross-section ($E_{\lambda 2}$) is also highly likely to be determined by the extracellular matrix (e.g., a stiffer ECM will offer a greater constraint to deformation).

An analysis of the difference in absolute values of $E_{\lambda 1}$ and $E_{max}$ at each ankle angle also showed that $E_{max}$ was greater than $E_{\lambda 1}$ in all three ankle angles but was significantly so only for the dorsiflexed ankle angle. In terms of the deformation pattern, this also indicates that while PF and N ankle angles are potentially between the schematics shown for asymmetric to highly asymmetric (Figure 2b,c), the dorsiflexion case may potentially correspond to highly asymmetric. Prior MR studies found a significant positive correlation of force in a cohort of young and old subjects or force loss due to unloading to the absolute value of the max shear strain ($E_{max}$) [7,20]. Thus, another potential reason for the higher force generated may arise for higher absolute values of $E_{max}$ in the dorsiflexed position.

A recent study measured intramuscular pressure (IMP) and EMG during isometric dorsiflexion (DF) MVC and isometric DF ramp contractions at DF, N, and plantarflexion (PF) ankle positions [21]. IMP was significantly correlated to the ankle torque during ramp contractions at each ankle position tested. However, the IMP did not reflect the change in the ankle torque which changed significantly at different ankle positions. Similar to the IMP study, the current study also showed that compressive strains at each ankle angle did not reflect the change in MVC at different ankle angles. However, normalized strains

(strains normalized to force) were significantly different between the ankle angles with an inverse correlation (higher force at DF was associated with the lowest normalized strain). An application of studying skeletal muscle under different ankle angle positions is in the examination of the EMG-torque slope in chronic stroke survivors [22]. The findings of the latter study suggest that muscular contraction efficiency is affected by hemispheric stroke, but in an angle-dependent and non-uniform manner. A future extension of the current work could be to study MRI-derived strain–force or strain–torque relationships in chronic stroke survivors to explore whether the patterns are similar to normal subjects or affected like the EMG-torque relationship [22]. It should also be noted that with the development of fast diffusion tensor imaging techniques such as the B-matrix spatial distribution method (BSD-DTI), it becomes more feasible to integrate dynamic strain mapping with diffusion tensor imaging [23].

There are some limitations to this study: (i) A single slice is acquired, limiting the strain analysis to a 2D tensor. In reality, the strain is a 3D tensor and volume imaging is required to capture the full trajectory of 3D tissue motion. However, it has been shown in earlier studies as well as in this study that the deformation is predominantly planar. It is demonstrated in this paper by the relatively small values of $E_{\text{out-plane}}$ or $SR_{\text{out-plane}}$. Further, care was taken to ensure that the acquired oblique sagittal slice captured the MG fibers in the imaging plane. (ii) The cohort size is small but the repeated measures design provides higher statistical power and statistically significant differences were seen between the normalized strain and strain rate values between the different ankle angles. This is a proof-of-concept paper where a new technique is established (2D strain tensor analysis to track changes of compressive, radial expansive and shear strains for different fiber architecture). The technique will be expanded in a future study to include a larger number of subjects and applied to studying differences with age and in disease conditions such as muscular dystrophy. The proposed MRI-based strain technique can also be adapted for elastography and compared to other techniques such as elastosonography [24].

## 5. Conclusions

In summary, 2D strain and strain rate tensor mapping of muscle deformation at three ankle angles provided insight on the effect of resting fiber length on the force generated. The decrease in MVC at the plantarflexed ankle angle position could tentatively be attributed to the shortened rest length which places it lower on the ascending arm of the FL curve. In addition to changes in contractile strain, this study revealed a strong asymmetry in deformation in the fiber cross-section with the highest asymmetry at the dorsiflexed ankle angle. The highest values of shear strain (relative to the contractile strain) were also seen at the dorsiflexed ankle angle. This study points to the potential contribution of factors besides the known contribution from the resting length of the muscle being close to optimum for force production in the dorsiflexed position. These additional factors are the increased deformation asymmetry and increased relative shear strain, representing two parameters that may also increase force production at the dorsiflexed position.

**Supplementary Materials:** The following supporting information can be downloaded at: https://www.mdpi.com/article/10.3390/tomography9020068/s1, Figure S1: Maps of the acquired velocity (phase) images and after denoising with 10 and 15 iterations respectively of the anisotropic diffusion filter. Figure S2: Maps of the positive strain ($E_{\lambda 2}$) and positive strain rate ($SR_{\lambda 2}$). Table S1: Strain rate indices for different ankle angles and %MVC. Table S2: Strain rate indices normalized to force for different ankle angles and %MVC.

**Author Contributions:** Conceptualization, U.S.; Formal analysis, R.H. and U.S.; Funding acquisition, S.S.; Methodology, V.M., B.C. and S.S.; Software, R.H. and V.M.; Supervision, U.S. and V.M.; Writing—original draft, U.S.; Writing—review and editing, R.H., U.S., V.M., B.C., E.S. and S.S. All authors have read and agreed to the published version of the manuscript.

**Funding:** This research was funded by National Institute on Aging (National Institute of Health, USA), grant number R01AG056999.

**Institutional Review Board Statement:** The study was conducted in accordance with the Declaration of Helsinki, and approved by the Institutional Review Board (or Ethics Committee) of University of California at San Diego (UCSD IRB# 171489, 6 July 2021) for studies involving humans.

**Informed Consent Statement:** Informed consent was obtained from all subjects involved in the study.

**Data Availability Statement:** The data presented in this study are available on request from the corresponding author.

**Conflicts of Interest:** The authors declare no conflict of interest. The funders had no role in the design of the study; in the collection, analyses, or interpretation of data; in the writing of the manuscript; or in the decision to publish the results.

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
