# Peer review of "Strain and Strain Rate Tensor Mapping of Medial Gastrocnemius at Submaximal Isometric Contraction and Three Ankle Angles"

_tomography, doi:10.3390/tomography9020068_

Round 1

Reviewer 1 Report

Dear Authors, congratulations for this wonderful work. Looking forward to reading your next article with diffusion data. Herein, I have some minor comments, please address them. 

1. Can you present phase images in without filter and with different k's in the supplementary. 

2. line 195: change the notation from k to K

3. Figure 2: in case 3:  is it Emax>Elambda1 or absolute of Emax> absolute of Elambda1

4. Figure 3. In the first frame, three red dots were seen. Are these representing the ROI in MG? if so, then please mention in the figure caption. 

5. Figure 4. it is hard to read numbers and text on the figure, thus it is difficult to understand anything from the figure. Please increase the resolution and font size. 

6. It is also interesting to see the difference between average strain and strain rates between proximal and middle ROIs.

7. The age ranges between subject are large as std is about 16. Thus, have you seen any difference between youngest and oldest subjects. Have you considered the inter-subject differences in the analysis?

8. Line 338: It is mentioned that Ex2 and Eout-planes are radial expansion, then why Eout-plane strain values are negative?  

9.Line 334: strain is subscripted, please correct it. 

10. Line 358: plantarflexed is subscripted, please correct it. 

11. Line 373: ankle is subscripted, please correct it. 

12. Line 376: largest differences values between E1 and E2 was seen PF. It should be DF ankle angle.

13. Line 387: position is subscripted, please correct it.

14. Line 385: is it maximum MVC or maximum strain?

15: Line 460-463: Rephrase sentence...

16: Line 471:"like the earlier studies" instead use "similar to previous studies."

17: Line 495: Can you elaborate what degree of strong anisotropic constrain requires raising force to double. 

18: Line 94: remove al from 3Dal.

Author Response

Detailed response to reviewers 1 and 2

Reviewer 2 Report

First of all, I would like to thank the authors of this paper. I found it to be quite an interesting scientific attempt and I always welcome papers, which are exploring new/better techniques for patient care…

When it comes to the review – the abstract should be totally modified according to scientific rules estimated as follows: introduction, material and methods, results and conclusions...

Authors have included only 6 (!) humans into the study, which is not valuable enough to prove their thesis. Work would be more valuable with larger sample size to validate the findings… This sample size is not applicable to be considered as a scientific research. 

Beginning of the manuscript starts with a profound introduction, which is too long - it should be shortened to the essential information regarding the work.

Authors estimated inclusion criteria for the study group -> which is "subject should be moderately active"... It's not a measurable criterion for the research! What does it mean to be moderately active?

Study was conducted in accordance with principles of the Declaration of Helsinki and informed consent was obtained from all the members of the study group. Authors have also provided us with the information that the study was approved by the Ethics Committee of University of California at San Diego. Authors have prepared statistical analysis with quite legible presentation of the results.  

Reference section consists of too old 23 positions. Some of them are dated back to 2006 or even to 1966 (!!!) - too outdated to be published. In the future, it would be better to moderate and expand the reference list with publications dating up to 5-6 years ago. It would improve the literature concerning the research.

Author Response

Detailed response to reviewers 1 and 2

Round 2

Reviewer 2 Report

-

Author Response

Response to the academic editor's comments

EdQ1:   We have added the statement on future work and comparison to elastosonography.

We have added this statement in the revision:

The cohort size is small but the repeated measures design provides higher statistical power and statistically significant differences were seen between the normalized strain and strain rate values between the different ankle angles. This is a proof-of-concept paper where a new technique is established (2D strain tensor analysis to track changes of compressive, radial expansive and shear strains for different fiber architecture). The technique will be expanded in a future study to include a larger number of subjects and applied to studying differences with age and in disease conditions such as muscular dystrophy. The proposed MRI based strain technique can also be adapted for elastography and compared to other techniques such as elastosonography [25].

EdQ2:   We have removed the two oldest references including the one from 1966. We have added five more recent references (all within the last four years) including the one mentioned here (the Mazur reference is Ref 24 in the revision).

We have added this in the revision:

It should also be noted that with the development of fast diffusion tensor imaging techniques such as the B-matrix spatial distribution method (BSD-DTI), it becomes more feasible to integrate dynamic strain mapping with diffusion tensor imaging [24]. 
